# A Logging Data Based Method for Evaluating the Fracability of a Gas Storage in Eastern China

Famu Huang [1], Lei Huang [2], Ziheng Zhu [1], Min Zhang [2,*], Wenpeng Zhang [1] and Xingwen Jiang [3]

[1] China Oil & Gas Pipeline Network Corporation, Shanghai 200092, China; huangfm@pipechina.com.cn (F.H.); zhuzh@pipechina.com.cn (Z.Z.); zhangwp05@pipechina.com.cn (W.Z.)
[2] State Key Laboratory of Marine Geology, Tongji University, Shanghai 201306, China; 2210858@tongji.edu.cn
[3] Key Laboratory of Continental Shale Hydrocarbon Accumulation and Efficient Development of Ministry of Education, Northeast Petroleum University, Daqing 163318, China; jcy102983452@163.com
* Correspondence: z_m@tongji.edu.cn

**Abstract:** Underground storage of natural gas has the characteristics of clean and low-carbon, and has the ability to provide a sustainable and stable supply. It is a very high-quality green energy that can increase the storage efficiency of gas storage through fracturing, achieving the sustainable development goal of "Carbon Peaking and Carbon Neutrality". To improve the storage efficiency of natural gas, it is necessary to carry out refracturing. Moreover, it is of great significance to estimate the fracability of the potential refracturing formation. At present, research on fracability is mainly based on qualitative characterization or quantitative evaluation based on rock mechanics and fracturing construction parameters, which cannot fully reflect the rock composition and structure of each stage. Firstly, based on logging data, this paper analyzes the evolution laws of strain energy such as elastic properties, pre-peak dissipation energy, and post-peak fracture energy during the transition of rock materials from plastic deformation to brittle fracture from an energy perspective, and determines the key energy that affects the brittle characteristics of rocks. Secondly, a brittleness index evaluation approach has been established that can comprehensively reflect the mechanical properties of rocks during pre-peak deformation and post-peak damage stages. In addition, this article focuses on the impact of a reservoir stratigraphic environment by combining the influence of geo-stresses with the rock brittleness index, and proposes a new method for evaluating reservoir fracability. Finally, this paper conducts a study on the fracability evaluation of three wells in a gas storage facility in Eastern China. The results indicate that low modulus and fracability index are beneficial for fracturing, thereby improving the gas production and peak shaving ability of gas storage.

**Keywords:** gas storage; energy method; brittleness index; fracability evaluation; fracturing

## 1. Introduction

Natural gas has the characteristics of clean and low-carbon, and the ability to provide a sustainable and stable supply, providing guarantees for achieving carbon peak and carbon neutrality, and is a sustainable green energy source. Underground energy storage is an efficient and environmentally friendly method of energy storage. Fracability refers to the ability of shale to undergo effective fracturing during the fracturing process, which determines the morphology of fractures and the complexity of fracture networks after fracturing. It is one of the important factors affecting the volume of reservoir reconstruction. Theoretical research has been conducted on the application of fracturing in underground gas storage [1–3]. Chevron has adopted a combined modification method based on hydraulic fracturing and nitrogen foam injection for gas reservoir modification technology. In this method, fracturing is utilized to break rocks and further enhance matrix conductivity, and nitrogen foam is injected at the same time to reduce the tension of the reservoir surface and improve the replacement effect. This method can improve the recovery rate and obtain economic benefits of gas reservoirs while reducing damage and pollution to the gas

reservoirs. Fu et al. (2017) investigated the influence of hydraulic fracturing on carbon storage performance [1]. Wang et al. (2022) proposed a mode containing multiple fractures; the results show that the degree of permeability anisotropy may change [2]. Xue et al. (2023) suggested a gas flow direction factor and transient heat transfer models for the alternative flow directions in the wellbore for first time fractured underground gas storage [3].

Fracability evaluation can be used in various underground engineering fields, such as gas storage construction, underground drainage construction, etc. [4–7]. Fracturing renovation of gas storage facilities can determine the stability and safe operating pressure boundary of the storage facility through the brittleness index. Repeated fracturing can also expand the storage capacity. Therefore, evaluating the compressibility of gas storage facilities is of great significance.

Foreign scholars were the first to use the brittleness index to characterize fracability, providing ideas for quantitative evaluation of fracability, but the research factors are relatively single [8–11]. Brittleness refers to a frame with minimal deformation, which refers to the deformation that an object can stand without losing its load-bearing capacity. Currently available rock brittleness evaluation methods include the following four main categories: (1) rock brittleness evaluation method based on mineral content [12,13]; (2) rock brittleness evaluation method based on logging data [13,14]; (3) rock brittleness evaluation method based on strength parameters [15–18]; (4) rock brittleness evaluation method based on strain [19–22]. Due to the lack of specific evaluation indicators and measurement methods in rock mechanics, scholars in different fields have proposed different definitions and calculation methods based on different evaluation objectives [23–26]. Mullen et al. (2012) and Jin et al. (2015) established different quantitative evaluation methods for fracability based on rock mechanics experiments and fracturing construction parameters [27,28]. Fracturing can increase oil and gas production and improve gas storage efficiency. Accurately evaluating the fracability of reservoirs is an important prerequisite for conducting reservoir fracturing design, which is of great significance for predicting the effectiveness of reservoir fracturing transformation, reasonably selecting fracturing well layers, and predicting post-fracturing production capacity [29–31]. At present, various fracturing models, established using various rock mechanics parameters, have been proven to be very effective methods [32,33]. Various evaluation methods require high reliability of parameters, so accurately obtaining reservoir rock mechanics parameters is crucial for conducting fracability evaluation [34]. In summary, the existing reservoir evaluation methods cannot fully reflect rock constitution structure at all stages, and the reservoir brittleness index alone cannot fully characterize the ease of hydraulic fracturing, and the formulation environment in which the rock is located also affects hydraulic fracturing. A good fracability model requires a balance between rock ontology and stratigraphic environment. We conduct research on the physical properties of rocks based on logging parameters, calculate the brittleness index of rocks using the energy method, and then comprehensively consider the coefficient of stress difference to evaluate the compressibility of reservoirs.

This paper analyzes the evolution law of strain energy, such as elastic properties, pre-peak dissipation energy, and post-peak fracture energy, during the transformation of rock materials from plastic deformation to brittle fracture from an energy perspective, and determines the key energy affecting the brittle characteristics of rocks, based upon which a brittleness evaluation index is established that can comprehensively reflect the mechanical characteristics of the pre-peak deformation and post-peak damage stages of rocks. In addition, this paper focuses on the influence of a reservoir stratigraphic environment, combines the ground stress influence with rock brittleness index, and proposes a new reservoir fracability evaluation method to comprehensively quantify natural properties, such as rock mechanical properties and stratigraphic environment, and provide reservoir data information and a theoretical basis for hydraulic fracturing design. Finally, a study on the fracability evaluation of three wells in a gas storage facility in Eastern China is evaluated, the research results indicate that the area has fracturing potential.

## 2. Fracability Evaluation Method

### 2.1. Energy Brittleness Index Method

The energy brittleness index used in this paper is based on the full stress–strain curve obtained from uniaxial compression experiments, which is divided into a pre-peak brittleness index and a post-peak brittleness index, with the breaking point as the dividing line. Since there are nonlinear segments in the full stress–strain curve, which is more complicated in the analysis process, the full stress–strain curve is linearly simplified, as shown in Figure 1.

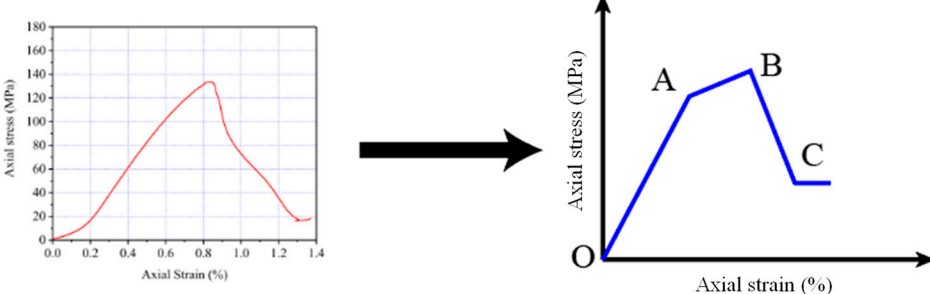

**Figure 1.** Simplified full stress–strain curve.

### 2.1.1. Pre-Peak Fragility Index

In conventional uniaxial compression experiments, the rock sample is first elastically deformed under the action of an axial load, and the external energy is accumulated inside the sample in the form of elastic energy. Figure 2 represents the elastic strain energy accumulated inside the rock sample during the elastic deformation phase of the rock. At this stage, if the external stress is withdrawn, the deformation of the rock will be fully recovered, and this part of the elastic strain energy will be fully released accordingly, at which time the slope of the straight line is the elastic modulus. Under the same deformation conditions, the larger the elastic modulus, the larger the elastic energy that the rock can accumulate. Therefore, the elastic modulus is not only a measure of the object's ability to resist elastic deformation, but also reflects the ability of the rock to accumulate energy.

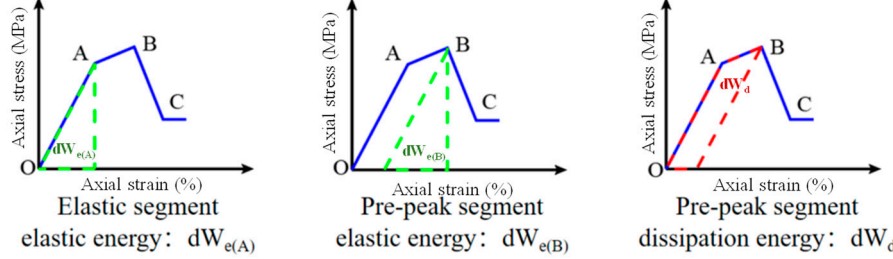

**Figure 2.** Schematic diagram of the energy evolution pattern inside the rock during the pre-peak deformation stage.

The above analysis shows that the pre-peak stage dissipation energy (dWd) has a significant effect on the rock brittleness, and the root reason is that the pre-peak dissipation energy is closely related to the "energy storage limit" of the rock itself. The rock's energy storage limit, i.e., the amount of energy accumulated at the peak of elastic strain energy, is used to characterize the rock's ability to accumulate elastic strain energy and is related to the nature of the rock itself and the stress state it is located in. When the elastic energy accumulated in the rock increases and reaches its energy storage limit, the energy accumulation effect stops and turns to release to the outside, and the rock will incur the overall fracture and rupture.

It is generally believed that the more energy consumed to make the rock break, the harder the rock is to break, the less brittle it is, and the more difficult it is to fracture

hydraulically. Applying this theory to the hydraulic fracturing process, for the purpose of starting a hydraulic fracture, the rock occurs elastic strain. At this time, the energy is concentrated in the rock, but this part of the energy is only stored in the rock; in the subsequent changes, the elastic energy in this part will also be released, so the greater the proportion of this part, which means the smaller the proportion of energy dissipated before the peak, the bigger the rock brittleness. When the rock is stressed to the yield point during hydraulic fracturing, micro-cracks and plastic strains are created inside the rock, and part of the energy is consumed, which cannot be recovered after consumption, so the larger the proportion of this part, the larger the proportion of dissipated energy before the peak, and the less the rock brittleness.

Due to the energy change, the elastic section is all elastic energy, and there is no energy dissipation, so the energy distribution before the peak occurs mainly in the plastic section. Part of the plastic section energy dissipates into the plastic section elastic energy and part dissipates into the plastic section dissipation energy. When the elastic section in the plastic section is the largest proportion of the total energy, the rock brittleness increases, and the plastic section total energy and plastic section elastic energy can be expressed as follows:

Plastic section elastic energy:

$$dW_{e(B)} - dW_{e(A)} \tag{1}$$

Plastic section total energy:

$$dW_d^* = dW_d + dW_{e(B)} - dW_{e(A)} \tag{2}$$

The pre-peak brittleness index can be represented by the ratio of the plastic section elastic energy to the plastic section total energy:

$$B_{pre-peak} = \frac{dW_d^*}{dW_{e(B)} - dW_{e(A)}} \tag{3}$$

The index shows that the closer the pre-peak brittleness index is to 1, the closer the total energy of the plastic section is to the elastic energy of the plastic section, and the smaller the plastic section dissipation energy, i.e., the more brittle the pre-peak section.

### 2.1.2. Post-Peak Brittleness Index

After reaching peak strength $\sigma_B$, the rock enters the fracture damage stage, where the microfractures inside the rock further expand and converge to form macroscopic fracture cracks, and the rock sample is completely damaged and loses a certain load-bearing capacity. Figure 3 shows the internal energy classification of the rock during the post-peak fracture phase.

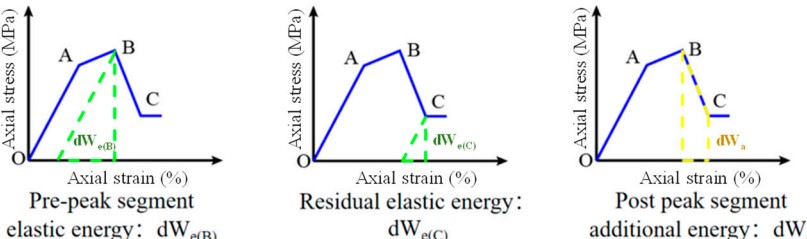

**Figure 3.** Schematic diagram of the energy evolution pattern during the post-peak stage.

Usually, the post-peak curve of the rock sample does not fall vertically but decreases gradually at a certain rate. This is due to the fact that after the peak strength is reached, the elastic energy stored inside the rock is not sufficient to sustain further fracture damage, and additional energy $(dW_a)$ is needed from outside to support this process. Under mechanical experimental conditions, this energy is partially provided by the continued loading of

the experimental machine. It can be seen that during the post-peak fracture phase, the following changes occurred within the rock: the elastic energy accumulated within the rock ($dW_{e(B)}$) and the additional energy provided by the testing machine ($dW_a$) together provide the energy for the fracture damage of the rock specimen. When the rock reaches the residual strength $\sigma_C$, due to its not completely lost load-bearing ability, there is still residual elastic energy inside the rock ($dW_{e(C)}$). The difference between $dW_{e(B)} + dW_a$ and $dW_{e(C)}$ is the fracture energy ($dW_F$) released by the rock fracture process. The fracture energy $dW_F$ is the key energy that determines the brittleness characteristics of rocks in the fracture damage stage. The smaller the fracture energy, the less extra energy the rock needs from the outside, the more violent the process of releasing energy from the rock, and the stronger the rock is in terms of brittleness. Bringing the above theory into hydraulic fracturing, it can be seen that when the reservoir rock is fractured, the elastic energy stored in the rock before the peak is released, and this part of the energy is used to continue to damage the rock to produce fractures, but the elastic energy is not enough to fully support the fracture extension process, and hydraulic fracturing still needs to continue to apply pump pressure to maintain fracture extension. And the lower the maintained pump pressure, the easier it is to complete fracturing, which means the stronger the rock brittleness. The fracture energy required for the fracture extension process of hydraulically fractured rocks can be divided into two parts: one part is the energy generated by additional pump pressure applied after the peak, and the other part is the elastic energy of the rock itself, released after the peak. The brittleness of the rock after the peak can be expressed by the ratio of the post-peak release elastic energy to the post-peak fracture energy, and the post-peak fracture energy to the post-peak release elastic energy can be expressed separately as:

Post-peak fracture energy:

$$dW_f = dW_a + \left( dW_{e(B)} - dW_{e(C)} \right) \tag{4}$$

Release of elastic energy after the peak:

$$dW_{e(B)} - dW_{e(C)} \tag{5}$$

The post-peak brittleness index can be represented by the ratio of the post-peak release elastic energy to the post-peak fracture energy:

$$B_{post-peak} = \frac{dW_f}{dW_{e(B)} - dW_{e(C)}} \tag{6}$$

The closer the post-peak brittleness index is to 1, the closer the post-peak dissipation energy is to the post-peak rock's own elastic energy consumption; that is, the release of elastic energy accumulated in the pre-peak section can complete most of the post-peak rock destruction process, and complete fragmentation can be achieved without applying additional energy, which means the post-peak section of the rock is more brittle.

### 2.1.3. Combined Brittleness Index

The combined brittleness index is derived from the combination of the pre-peak brittleness index and the post-peak brittleness index, as shown in the following equation:

$$B = B_{pre-peak} \times B_{post-peak} = \frac{dW_d^*}{dW_{e(B)} - dW_{e(A)}} \times \frac{dW_f}{dW_{e(B)} - dW_{e(C)}} \tag{7}$$

The above equation shows that the pre-peak brittleness index and post-peak brittleness index are both 1 for completely brittle rocks, so the neutralizing brittleness index for completely brittle rocks should also be 1. The larger the value, the weaker the brittleness.

The comprehensive brittleness index is calculated by strain energy, but the practical application of calculating strain energy is difficult, so the comprehensive brittleness index

is simplified. The comprehensive brittleness index is proposed based on the uniaxial all-stress–strain curve, which has been simplified in Figure 1 by linearizing the all-stress–strain curve and splitting the whole curve into three linear segments, namely, the elastic segment, the plastic segment, and the post-peak segment. Li et al. (2019) have conducted related experiments based on the energy method to calculate the brittleness of coal, verifying the feasibility of the method from an experimental perspective [35]. The deformation and failure of rocks under external loads can be divided into three stages based on the energy method. The first stage is the energy accumulation stage. This stage mainly focuses on the transformation of external load work and rock elastic performance. The second stage is the energy dissipation stage, roughly corresponding to the unstable fracture stage, which mainly involves the conversion of elastic energy and damage dissipation energy. The third stage is the energy release stage, corresponding to the post-peak softening stage, during which a large amount of elastic energy is released and converted into surface energy and kinetic energy of the fragments. The slope of these three linear segments is calculated, and the slope of the elastic segment is the elastic modulus (E), the slope of the plastic segment is the yield modulus (D), and the slope of the post-peak segment is the post-peak modulus (M), as shown in Figure 4.

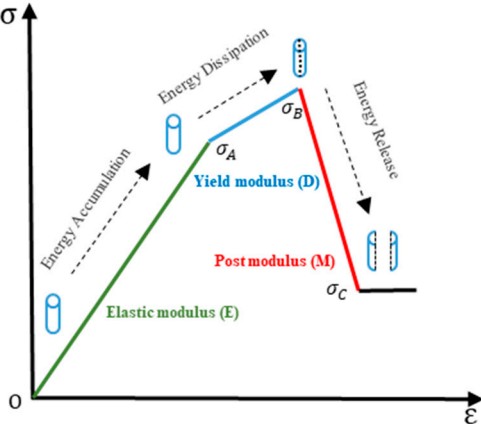

**Figure 4.** Full stress–strain curve in three-modulus form.

The elastic modulus, yield modulus, and post-peak modulus are relatively easier to calculate and more intuitive, so this project simplifies the combined brittleness index by three types of modulus.

For the pre-peak index, it is obtained that

$$dW_d^* = \frac{\sigma_B^2 - \sigma_A^2}{2D} \tag{8}$$

$$dW_{e(B)} - dW_{e(C)} = \frac{\sigma_B^2 - \sigma_C^2}{2E} \tag{9}$$

For the post-peak index, it can be obtained from

$$dW_f = dW_{e(B)} + dW_a - dW_{e(C)} = \frac{\sigma_B^2}{2E} + \frac{\sigma_B^2 - \sigma_C^2}{-2M} - \frac{\sigma_C^2}{2E} = \frac{\left(\sigma_B^2 - \sigma_C^2\right)(M - E)}{2ME} \tag{10}$$

$$dW_{e(B)} - dW_{e(A)} = \frac{\sigma_B^2 - \sigma_A^2}{2E} \tag{11}$$

Therefore, the combined fragility index is

$$B = \frac{dW_d^*}{dW_{e(B)} - dW_{e(A)}} \times \frac{dW_f}{dW_{e(B)} - dW_{e(C)}} = \frac{E}{D} \times \frac{M - E}{M} \tag{12}$$

2.1.4. Calculation of Brittleness Index from Logging Data

The integrated brittleness index in Equation (12) shows that to obtain the brittleness index, only the elastic modulus, yield modulus, and post-peak modulus need to be calculated. In this section, the correlation between the logging data and the three modulus quantities is established separately along the above lines, and then the brittleness index is calculated.

The elastic modulus can be divided into a dynamic elastic modulus and a static elastic modulus. Since the elastic modulus used in the energy method is obtained in the full stress–strain curve, the elastic modulus is the static elastic modulus, which can be obtained by the dynamic elastic modulus, and the dynamic elastic modulus can be obtained by the transverse and longitudinal wave acoustic time difference and density.

The dynamic elastic modulus is calculated using the following equation:

$$E_d = \frac{ZDEN \times \left(3 \times DTS^2 - 4 \times DTC^2\right)}{DTS^2 \left(DTS^2 - DTC^2\right)} \times 9.299 \times 10^7 \tag{13}$$

where, $E_d$ is the dynamic modulus of elasticity, MPa; $ZDEN$ is the density, g/cm$^3$; $DTC$ is the longitudinal acoustic time difference, μs/ft; and $DTS$ is the transverse acoustic time difference, μs/ft.

The above formula is used to calculate the dynamic modulus of elasticity, and the static modulus of elasticity can be calculated by Equation (14):

$$E_s = 0.18945E_d + 5.66963 \tag{14}$$

The plastic modulus is less often used in practical engineering, so there are no mature conversion equations, as for the elastic modulus. Therefore, we solve them based on the relevant parameters of the well logging curve through the stress–strain curve. For the plastic modulus, we need to solve for the peak stress and its corresponding strain value. Correspondingly, for the weakened modulus, we need to solve for the yield stress and its corresponding strain value. The solution of these stress and strain values can be derived through logging parameters and related physical models.

Firstly, the rock mechanics parameters are calculated based on the logging curve, and the main solving parameters are as follows:

(1)     Velocity conversion of longitudinal and transverse sound waves

$$V_p = 304.8 \times \frac{1}{\Delta t} \tag{15}$$

where $V_p$ is the longitudinal wave velocity, km/s; $\Delta t$ is measured acoustic time difference, μs/ft.

(2)     Effective stress coefficient (Biot's coefficient)

$$\alpha = 1 - \frac{\rho \left(3V_p^2 - 4V_s^2\right)}{\rho_m \left(3V_{mp}^2 - 4V_{ms}^2\right)} \tag{16}$$

where $\rho$ is the density value of the formation, g/cm$^3$; $\rho_m$ is the density of the skeleton rock material, g/cm$^3$, taken from dense sandstone $\rho_m = 2.65$, input from other lithology; $V_{mp}$ is the longitudinal wave velocity of the skeleton material, km/s, and $V_{mp}$ = 5.95 or dense

sandstone 95, artificial input from other lithologies; $V_{ms}$ is the shear wave velocity of the skeleton material, km/s, and $V_{ms} = 3.0$ is taken for dense sandstone, with input from other lithologies; $V_p$ is the longitudinal wave velocity of the formation in km/s; Vs is the shear wave velocity of the formation in km/s.

(3)　　Mud content

Using gamma data, the mud content is calculated by calculating the mud content calculation formula as:

$$I_{sh} = \frac{2^{\Delta GR \times G_{CUR}} - 1}{2^{G_{CUR}} - 1} \tag{17}$$

$$\Delta GR = \frac{GR - GR_{min}}{GR_{max} - GR_{min}} \tag{18}$$

where $I_{sh}$ is the mud mass fraction; $\Delta GR$ is the natural gamma difference; $GR_{max}$ and $GR_{min}$ are the maximum and minimum values of natural gamma in the logging curve, GAPI, respectively; and $G_{CUR}$ is the formation age correction factor, 3.7 for new formations and 2.0 for old formations.

(4)　　Uniaxial tensile strength of rocks

The uniaxial compressive strength is:

$$S_c = 0.033\rho^2 V_p^4 \left(\frac{1 + \mu_d}{1 - \mu_d}\right)^2 (1 - 2\mu_d)(1 + 0.78I_{sh}) \tag{19}$$

$$S_t = \frac{S_c}{K} \tag{20}$$

where $S_c$ is the uniaxial compressive strength, MPa; $\rho$ is the rock mass density, g/cm$^3$; $\mu_d$ is the dynamic Poisson's ratio, dimensionless; $S_t$ is the uniaxial tensile strength, MPa; The commonly used range of K values is 8–25, with a temporary value of 12.

(5)　　Formation pore pressure

$$P_p = DEPT \times 1.2/100 \tag{21}$$

where *DEPT* is the depth of the well, m.

(6)　　Vertical stress and maximum and minimum horizontal principal stresses

The maximum horizontal ground stress and the minimum horizontal ground stress need to be calculated when calculating the ground stress difference coefficient. There are many methods to calculate horizontal ground stress, among which, Huang's model is the most widely used. In this paper, Huang's model is used for calculation, and the specific formula is as follows:

$$\sigma_H = [\frac{\nu}{1 - \nu} + A](\sigma_v - VP_P) + VP_P \tag{22}$$

$$\sigma_h = [\frac{\nu}{1 - \nu} + B](\sigma_v - VP_P) + VP_P \tag{23}$$

$$\sigma_v = g\int_0^D \rho_b(h)dh \tag{24}$$

$$\nu = \frac{DTS^2 - 2DTC^2}{2(DTS^2 - DTC^2)} \tag{25}$$

where, $\sigma_v$ is vertical stress, MPa; $\rho_b$ is density, g/cm$^3$; $\nu$ is Poisson's ratio, which can be calculated by Equation (27); *DEPT* is well depth, m; $P_P$ is pore pressure, MPa; *V* is effective stress coefficient, which is taken as 0.8, according to the data; A and B are tectonic coefficients, which are taken as 0.575 and 0.315, respectively, in this block.

By solving the above parameters, we can further calculate the required peak stress and yield stress. The peak stress is the highest point that appears on the stress–strain curve, also known as peak strength. The peak strength is calculated using the mud content and dynamic modulus of elasticity by using the compressive strength formula, which is calculated as:

$$\sigma_b = (0.0045 + 0.0035 I_{sh}) E_d \tag{26}$$

where $\sigma_b$ is the uniaxial compressive strength, MPa.

The yield stress is the strength value at which a rock ruptures, and at this point, the stress does not significantly change with strain. Here, we use fracture pressure to approximate yield stress, and the specific solution for formation fracture pressure is as follows:

$$\sigma_c = P_f = 3\sigma_h - \sigma_H - \alpha P_p + S_t \tag{27}$$

where $\sigma_c$ is the yield stress, MPa; $P_f$ is the formation fracture pressure, MPa; $\sigma_H$ is the maximum horizontal ground stress, MPa; $\sigma_h$ is the minimum horizontal ground stress, MPa; $P_p$ is the formation pore pressure, MPa; $\alpha$ is the effective stress coefficient, dimensionless; $S_t$ is the uniaxial tensile strength, MPa.

For the solution of strain values, the constitutive equation is obtained according to Lemaitre's strain equivalence principle [36,37]:

$$\varepsilon_i = \frac{1}{E}[\sigma_i - \mu(\sigma_j + \sigma_k)] \tag{28}$$

where $\varepsilon_i$ is the strain value in the i direction; E is the elastic modulus; $\mu$ is Poisson's ratio; $\sigma_i$, $\sigma_j$, $\sigma_k$ is the stress in the i, j, and k directions, respectively.

Assuming that the peak stress and yield stress are in the k direction, under tensile/compressive stress, the maximum horizontal principal stress is perpendicular to the tensile/compressive direction, and the minimum horizontal principal stress is parallel to the tensile/compressive direction. We use the maximum horizontal principal stress and the minimum horizontal principal stress to replace the stress values in the i and j directions. Therefore, the calculation results of strain values corresponding to peak stress and yield stress are as follows:

$$\varepsilon_b = \frac{1}{E}[\sigma_b - \mu(\sigma_h + \sigma_H)] \tag{29}$$

$$\varepsilon_c = \frac{1}{E}[\sigma_c - \mu(\sigma_h + \sigma_H)] \tag{30}$$

By combining the strain values, we can calculate the yield modulus *D* and post-peak modulus *M* at each point in the logging data:

$$D = \frac{\sigma_b - E}{\varepsilon_{i\_b} - 1} \tag{31}$$

$$M = \frac{\sigma_b - \sigma_c}{\varepsilon_{i\_c} - 1} \tag{32}$$

## 2.2. Brittle Ground Stress Fracability Index

In the previous section, the energy method brittleness index is proposed to characterize the nature of the reservoir rock itself, but in the actual hydraulic fracturing process, the single property of rock brittleness alone cannot determine the ease of hydraulic fracturing, and the ground stress also affects the hydraulic fracturing process, in which the reservoir rock is more likely to fracture the rock at high ground-stress difference, while the fracture initiation and extension is much more difficult at low ground-stress difference than at high ground-stress difference. The ground-stress condition and rock brittleness are both natural properties, independent of hydraulic fracturing design, and their values directly affect the fracturing difficulty and results. Therefore, the evaluation of natural reservoir fracability

needs to take into account both the ground stress condition and reservoir rock brittleness, for which a new fracability index is developed based on the energy method brittleness index in the following form:

$$F_{new} = B_n \times \Delta\sigma_n \tag{33}$$

$$B_n = \left(B - B_{(min)}\right) / \left(B_{(max)} - B_{(min)}\right) \tag{34}$$

$$\Delta\sigma_n = \left(\Delta\sigma_{(min)} - \Delta\sigma\right) / \left(\Delta\sigma_{(min)} - \Delta\sigma_{(max)}\right) \tag{35}$$

$$\Delta\sigma = \frac{\sigma_H - \sigma_h}{\sigma_h} \tag{36}$$

where $\Delta\sigma$ is the ground stress difference coefficient; $B_n$ and $\Delta\sigma_n$ are the normalized results of the brittleness index and the ground stress difference coefficient.

As shown in Equation (33), the new fracability index multiplies the brittleness index and the ground stress discrepancy coefficient, allowing them to jointly influence the fracability index magnitude. For example, when the rock is more brittle but the ground stress difference is very small, fracturing is more difficult and the corresponding fracability index is smaller.

Thus, the elastic modulus, yield modulus, and post-peak modulus required to calculate the energy brittleness index are found. The types of logging data required are density, interval transit time, natural gamma, and depth. The dynamic elastic modulus is calculated using the density and transverse and longitudinal acoustic time difference; the static elastic modulus is calculated using the dynamic and static elastic modulus conversion equation. The yield modulus is calculated using peak strength and corresponding strain. The post-peak modulus is calculated using fracture pressure and corresponding strain. Then, the brittleness index is calculated from static modulus of elasticity, yield modulus, and post-peak modulus. Finally, combining the stress difference coefficient, the fracability evaluation index is obtained. The calculation flow of the fracability evaluation index is show in Figure 5.

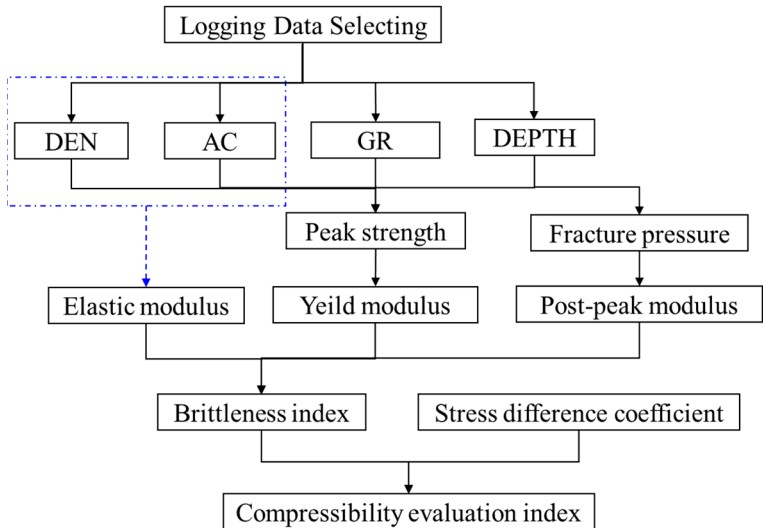

**Figure 5.** Calculation flow of fracability evaluation index by energy method.

## 3. Results and Discussion

Using relevant logging data from three wells in a gas storage in Eastern China, the depth range of the wells is 1600–3200 m. The relevant logging parameters include density value, acoustic time difference, natural gamma value, etc. Based on the mathematical model established in this paper, to calculate fracability index, relevant research is conducted. The

well map of the gas storage is shown in Figure 6, where XX-1, XX-2, and XX-3 represent the three wells calculated in our model.

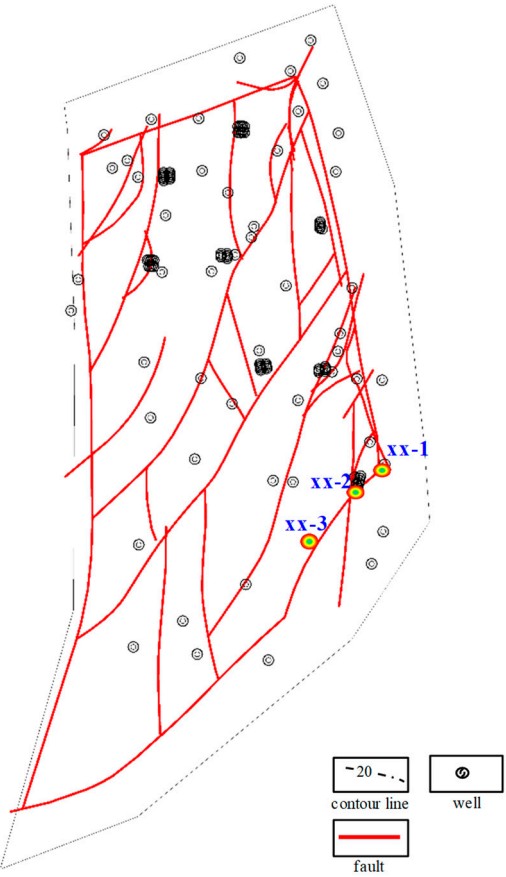

**Figure 6.** Location well map of a gas storage in Eastern China.

Figure 7 shows the relationship between elastic/yield/post-peak modulus and well depth. It can be seen that when the well depth range is 2000–3200 m, the elastic modulus is mainly distributed between 10 GPa and 40 GPa, the plastic modulus is mainly distributed between 5 GPa and 30 GPa, and the post-peak modulus is mainly distributed between 10 GPa and 60 GPa. The larger the modulus of a rock, the smaller the strain. Therefore, the modulus is usually used to reflect the ability of shale to maintain fractures after fracturing. The higher the modulus, the stronger its brittleness. The mean values of elastic modulus, yield modulus and post-peak modulus decrease with the increase in well depth, indicating that the brittleness of the reservoir weakens with the increase in well depth.

Figure 8 shows the relationship between the fracability index and well depth. It can be seen that the fracability index is mainly concentrated between 0.45 and 0.65, and decreases with the increase in well depth. As the well depth increases, the modulus of the reservoir decreases relatively, the brittleness weakens, and the fracability decreases. Strata with strong brittleness are sensitive to fracturing operations, with a large renovation area and good communication with natural fractures. The fracture network can effectively spread and form a complex fracture network, improving single-well production. On the contrary, if the brittleness of the reservoir rock is poor and the toughness is high, the reservoir rock undergoes compression deformation, resulting in poor fracture effect and ineffective extension of the main and branch fractures, resulting in a poor overall transformation effect. On the other hand, the deeper the reservoir, the greater the overlying stress, which is less conducive to the development of fracturing. Furthermore, the deeper the reservoir, the greater the coefficient of horizontal stress difference, and the hydraulic fractures propagate in a single direction, which is more unfavorable for the formation of fracture networks and weakens their fracability.

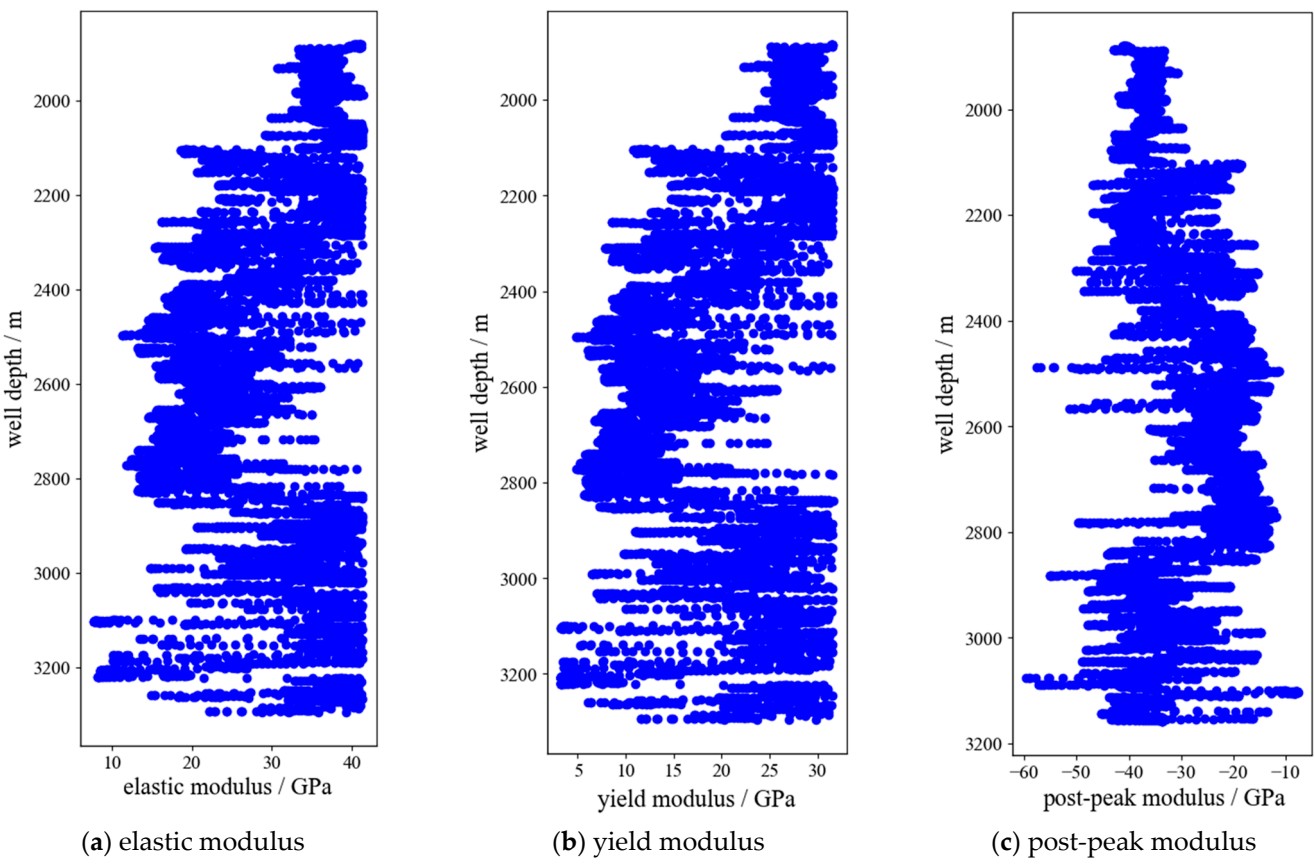

(**a**) elastic modulus                    (**b**) yield modulus                    (**c**) post-peak modulus

**Figure 7.** The relationship between the elastic/yield/post-peak modulus and well depth.

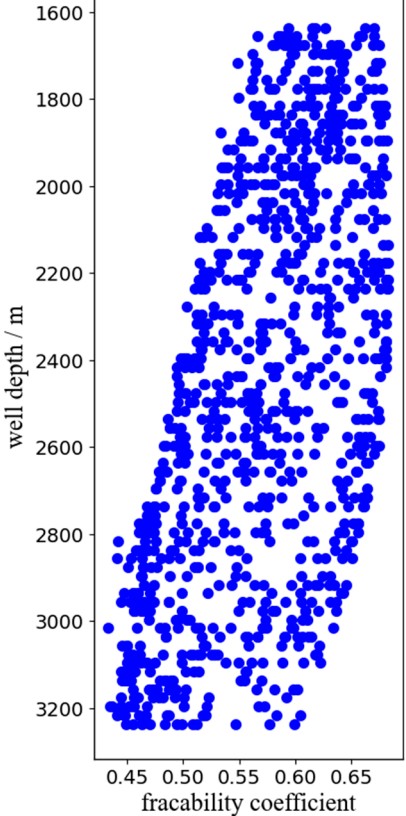

**Figure 8.** The relationship between the fracability index and well depth.

Figure 9 shows the magnitude of fracability index under Kriging three-dimensional interpolation, showing the distribution of fracability in the three-dimensional space of the reservoir, and also showing the well trajectories of three wells. From the figure, it can be seen that the fracability index of the reservoir within the depth range of 1600–3200 m is mainly concentrated between 0.45 and 0.65, indicating that the reservoir where these three wells are located is suitable for fracturing. To better demonstrate the distribution of fracability on each surface, contour plots were performed on each layer, as shown in Figures 10–12.

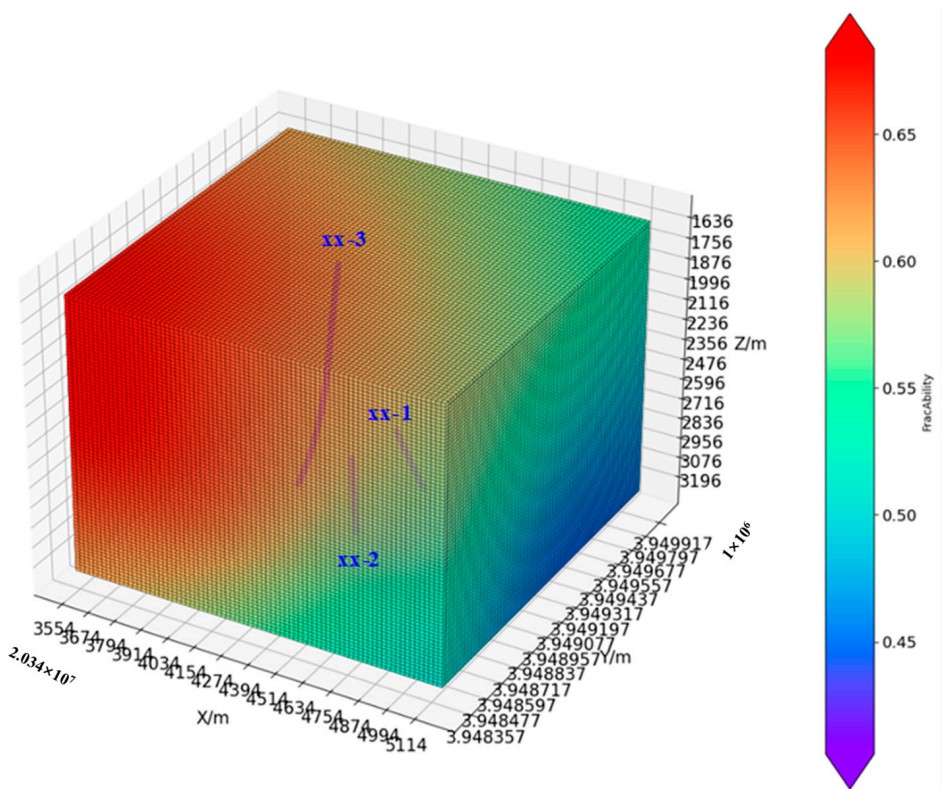

**Figure 9.** Schematic diagram of fracability index under three-dimensional Kriging interpolation.

Figure 10 shows the distribution of fracability in the X direction, with values of 20,344,300 m, 20,344,350 m, 20,344,350 m, 2,034,400 m and 20,344,450 m, respectively. It can be seen that the fracability coefficients of wells XX-1 and XX-2 are mainly concentrated between 0.40 and 0.45, while the fracability coefficients of wells XX-3 are mainly concentrated between 0.45 and 0.65. As the X value increases, the overall fracability index of the YZ plane decreases, indicating a decreasing trend in the X direction.

Figure 11 shows the fracability index in the Y direction, with Y values of 39,489,000 m/ 39,489,100 m/39,489,200 m/394,894,400 m. As the Y value increases, the fracability of the XZ plane does not change significantly, and its value decreases from southwest to northeast. This indicates that the fracability distribution is relatively uniform in the XZ plane and decreases from southwest to northeast.

Figure 12 shows the distribution of fracability in the Z direction, with depths of 2400 m, 2600 m, 2800 m, 3000 m, respectively. It can be seen that as the Z value increases, the fracability index significantly decreases and the inter-well interference relatively weakens. According to the fracability contour map of the reservoir in the X, Y, and Z directions, the fracability index is uniform in the XZ plane, but non-uniform in the XY and YZ planes. As the X and Z values increase, the fracability decreases.

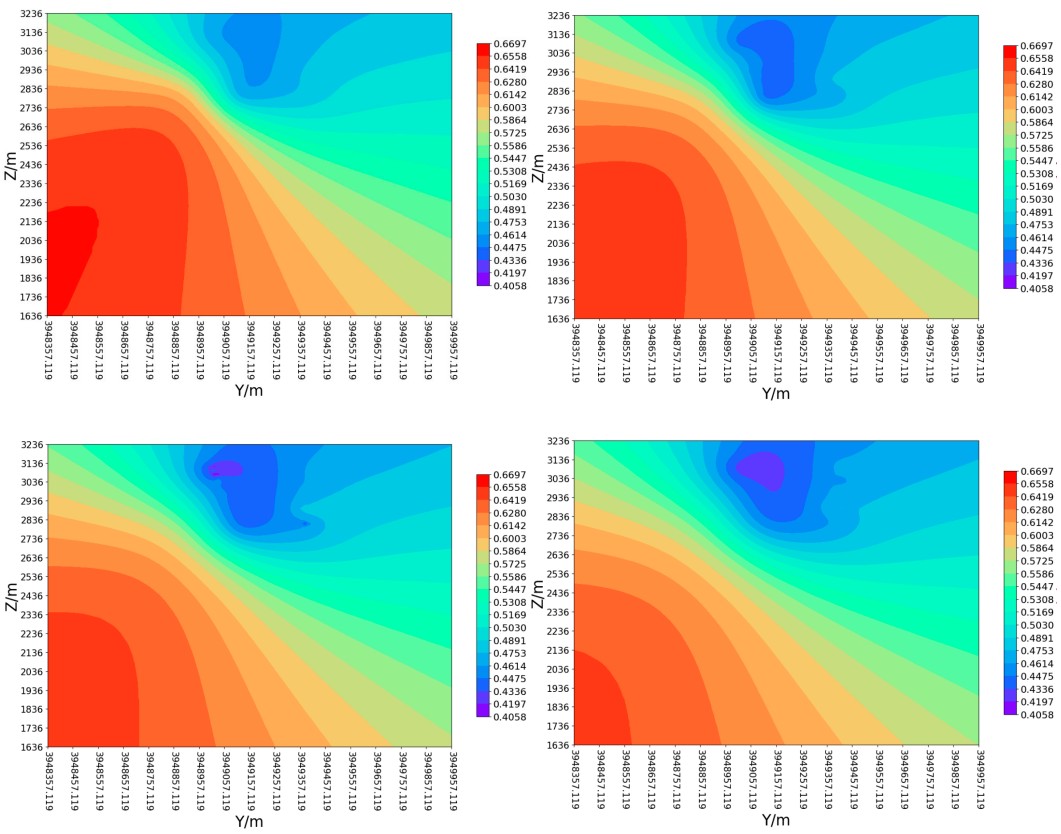

**Figure 10.** Contour plot of fracability index in the X-direction. (X = 20,344,300 m/20,344,350 m/ 2,034,400 m/20,344,450 m).

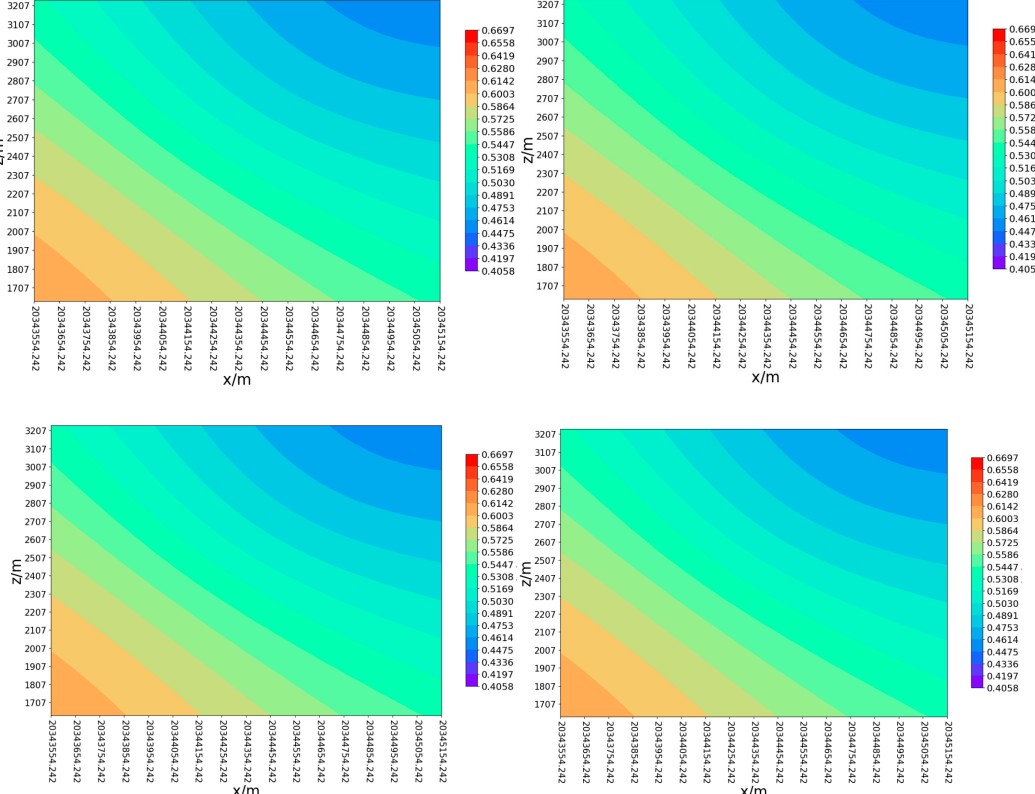

**Figure 11.** Contour plot of fracability index in the Y-direction. (Y = 39,489,000 m/39,489,100 m/ 39,489,200 m/39,489,400 m).

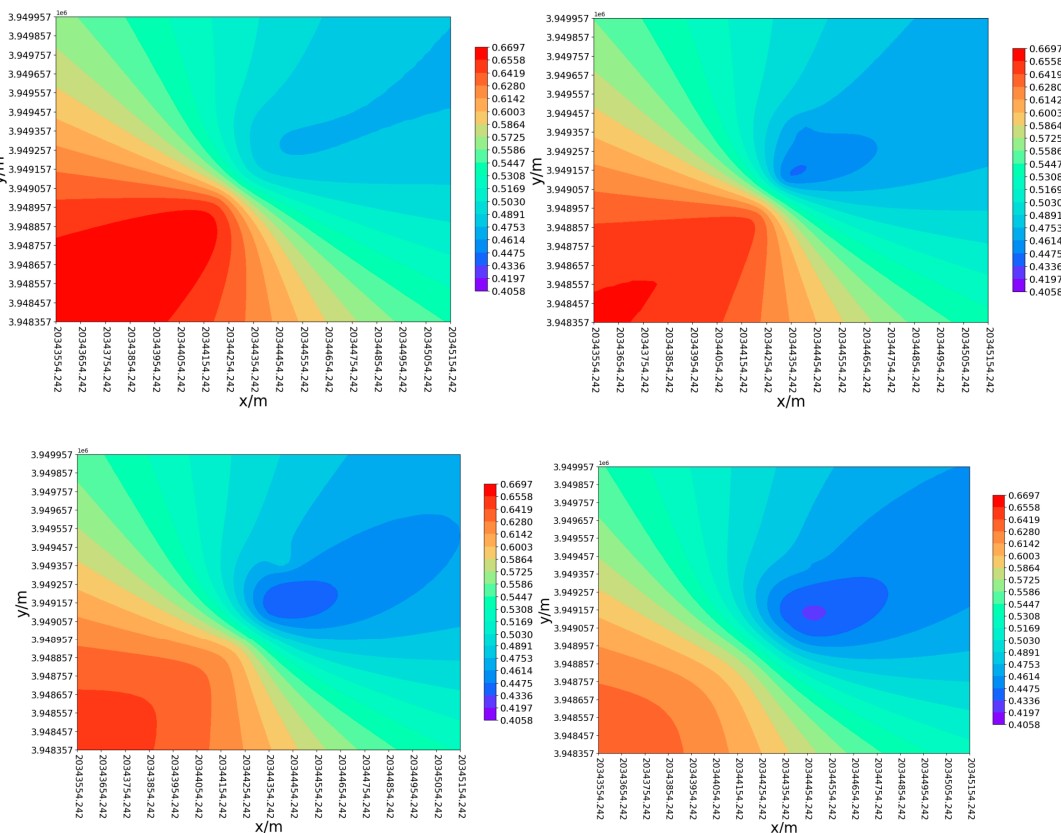

**Figure 12.** Contour plot of fracability index in the Z direction. (Z = 2400 m/2600 m /2800 m/3000 m).

Figure 13 shows the variation of relevant parameters of the XX-2 well with well depth, mainly including rock fracability coefficient, elastic modulus, Poisson's ratio, and shale content. The elastic modulus is mainly distributed between 16.16 GPa and 47.02 GPa; the Poisson's ratio is mainly concentrated between 0.111 and 0.163; the mud content is mainly concentrated between 0.018 and 0.324; and the fracability index is mainly distributed around 0.45. The smaller the Poisson's ratio of a rock, the smaller its deformation capacity before fracture. Therefore, the Poisson's ratio reflects the ability of a rock to fracture under a certain pressure. It is generally believed that rocks with low Poisson's ratio and high Young's modulus have higher brittleness.

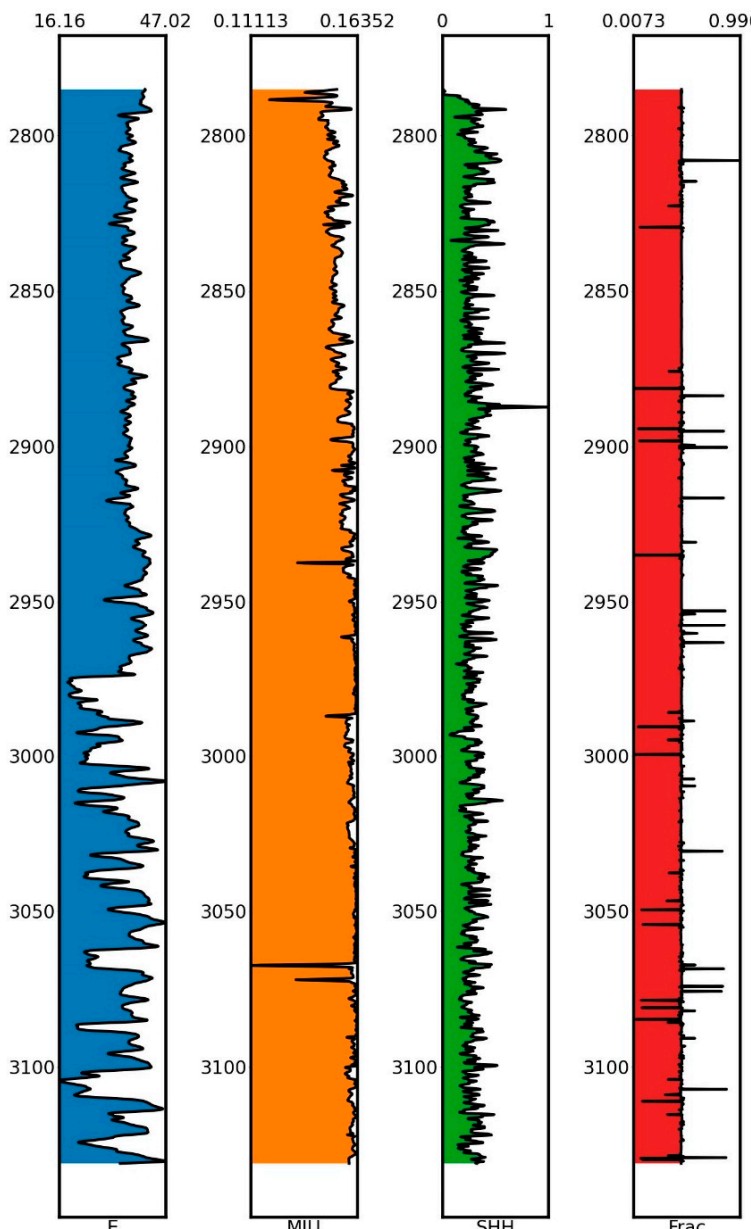

**Figure 13.** Schematic diagram of fracability index (FRAC), elastic modulus (E), Poisson's ratio (MIU), and shale content (SHH) changing with well depth.

### 4. Conclusions

Based on the well logging data, this paper proposes a brittleness evaluation index that can comprehensively reflect the mechanical properties of rocks during the pre-peak deformation stage and the post-peak damage stage. A new fracability evaluation method that comprehensively considers the influence of in-situ stress and rock brittleness index is proposed. Finally, a fracability investigation on the three wells of a gas storage in Eastern China is conducted, and the main research conclusions are shown as follows:

(1) As the formation depth increases, the elastic modulus, yield modulus, and post-peak modulus decrease, resulting in a decrement of reservoir brittleness and fracability, which is more unfavorable for the refracturing of underground gas storage.

(2) With the increment of formation depth, the fracability index decreases. The fracability index mainly stays within the range from 0.45 to 0.65, which indicates that the overall reservoir in this area has fracturing potential.

(3) By calculating the fracability index based on three-dimensional Kriging interpolation, it can be seen from the fracability contour map in the X, Y, and Z directions that the fracability index is uniformly distributed in the XZ plane but non-uniformly distributed in the XY and YZ planes. Moreover, the fracability index has a negative correlation with the X and Z values.

(4) Based on the well logging data and calculation results of rock physical parameters related to the XX-2 well, it can be concluded that its elastic modulus primarily ranges from 16.16 GPa to 47.02 GPa, and the Poisson's ratio is mainly concentrated between 0.111 and 0.163. In addition, the mud content is mainly concentrated between 0.018 and 0.324, and the fracability index is mainly distributed around 0.45.

**Author Contributions:** Conceptualization, F.H. and L.H.; Methodology, M.Z.; Validation, Z.Z., W.Z. and X.J.; Investigation, X.J.; Data curation, W.Z.; Writing—original draft, L.H. and M.Z.; Writing—review & editing, F.H.; Supervision, Z.Z. All authors have read and agreed to the published version of the manuscript.

**Funding:** This research received no external funding.

**Institutional Review Board Statement:** Not applicable.

**Informed Consent Statement:** Not applicable.

**Data Availability Statement:** No new data were created or analyzed in this study. Data sharing is not applicable to this article.

**Conflicts of Interest:** Author Mr. Famu Huang, Mr. Ziheng Zhu and Mr. Wenpeng Zhang are employed by the China Oil & Gas Pipeline Network Corporation. The remaining authors declare that the research was conducted in the absence of any commercial or financial relationships that could be construed as a potential conflict of interest.

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
