# Peer review of "A Logging Data Based Method for Evaluating the Fracability of a Gas Storage in Eastern China"

_sustainability, doi:10.3390/su16083165_

Round 1

Reviewer 1 Report

Comments and Suggestions for Authors

In this paper, the authors propose a new method for evaluating reservoir fracturing characteristics and introduce a brittleness assessment index that reflects the mechanical properties of pre-peak deformation and post-peak damage stages of the rock. This method is applied to evaluate the fracturing characteristics of three wells in a gas storage facility in eastern China. The authors combine the influence of in-situ stress with the rock brittleness index to comprehensively quantify the rock's mechanical properties and natural attributes of the formation environment. The results can provide reservoir data and theoretical basis for hydraulic fracturing design. However, further improvement is needed regarding the theoretical feasibility of the proposed method and the logical description of the obtained results. The following are some additional detailed comments for the authors' consideration.

1) In the introduction section, there have been provided a comprehensive overview of scholars' research on rock fracturing characteristics and rock brittleness, both domestically and internationally. I would like to kindly request for a more detailed description of your specific contributions and innovative aspects in this field.

2) In Section2.1 the Energy Brittleness Index Method, authors propose a simplification approach for the actual model curve and subsequently established a series of evaluation indicators. Please provide more information on whether there are precedents for this simplification method and if the evaluation indicators derived from it can be applied to practical scenarios? Please kindly provide further details and clarification.

3) In Figure 4, the division of the curve into three parts in the Full stress-strain curve in three modulus form appears to be hasty. It is recommended that the authors provide additional explanations and evidence regarding the feasibility of this theoretical approach.

4) The lines in Figure 5 are overlapping and appear to be visually confusing, which makes the logic unclear. I suggest redrawing the figure to improve its clarity.

5) The relationship between the content presented in Figure 7 and the accompanying text stating ‘Figure 7 shows the relationship between elastic/yield/post-peak modulus and well 349 depth...’ (lines 347-358) is not clearly evident. It is recommended that the authors revise and refine this section to ensure better alignment between the visual content and the accompanying text.

6) Some related papers (e.g., Journal of Cleaner Production 354 (2022) 131724) are suggested to cited.

7) Many grammar issues in the text need to be corrected. For example, in the fourth part, ‘The key energy’ (lines 433) should be corrected. The sentence ‘which is more unfavorable for reservoir fracturing’ (lines 442) is grammatically correct and there are no language errors. However, it may be unclear what "which" is referring to without the context of the previous sentence.

Comments on the Quality of English Language

NA

Author Response

Responses to Technical Editor 1: 

In this paper, the authors propose a new method for evaluating reservoir fracturing characteristics and introduce a brittleness assessment index that reflects the mechanical properties of pre-peak deformation and post-peak damage stages of the rock. This method is applied to evaluate the fracturing characteristics of three wells in a gas storage facility in eastern China. The authors combine the influence of in-situ stress with the rock brittleness index to comprehensively quantify the rock's mechanical properties and natural attributes of the formation environment. The results can provide reservoir data and theoretical basis for hydraulic fracturing design. However, further improvement is needed regarding the theoretical feasibility of the proposed method and the logical description of the obtained results. The following are some additional detailed comments for the authors' consideration.

Thank you very much for your kind review. We have responded point-by-point according to your comments.

[Comment 1]: In the introduction section, there have been provided a comprehensive overview of scholars' research on rock fracturing characteristics and rock brittleness, both domestically and internationally. I would like to kindly request for a more detailed description of your specific contributions and innovative aspects in this field.

 [Reply 1]: Thanks for your kind comments. We conduct research on the physical properties of rocks based on logging parameters, calculate the brittleness index of rocks using the energy method, and then comprehensively consider the coefficient of stress difference to evaluate the fracability of reservoirs. The innovation of this paper lies in the use of fracability evaluation in the construction of gas storage. Through fracability evaluation, we can determine the stability of the gas storage reservoir, and repeated fracturing can expand its capacity and increase its storage efficiency.

In the revised manuscript, special contributions and related innovations have been added in the introduction section.

[Changes]: Please see Line 43-59 on Page 2; Line 79-88 on Page 3 in the "Tracked changes file".

    [Comment 2]: In Section ‘2.1 the Energy Brittleness Index Method’, authors propose a simplification approach for the actual model curve and subsequently established a series of evaluation indicators. Please provide more information on whether there are precedents for this simplification method and if the evaluation indicators derived from it can be applied to practical scenarios? Please kindly provide further details and clarification.

 [Reply 2]: Thank you for your useful advice. Fig.1 shows a graph of the relationship between rock energy evolution and stress-strain. The OA segment represents the accumulation of energy, the AB segment represents the dissipation of energy, and the BC segment represents the release of energy. This graph is a typical stress-strain curve in rock mechanics, and scholars at home and abroad have adopted this simplified model [1-3].

References:

[1] Zhang, J.;  Ai, C.; Li, Y. et al. Energy-Based Brittleness Index and Acoustic Emission Characteristics of Anisotropic Coal Under Triaxial Stress Condition. Rock Mechanics and Rock Engineering. 2018, 51: 3343-3360.

[2]  Ai, C.; Zhang, J.;  Li, Y. et al. Estimation Criteria for Rock Brittleness Based on Energy Analysis During the Rupturing Process. Rock Mechanics and Rock Engineering. 2016, 49: 4681-4698.

[3] Li, Y.; Zhou, L.; Li, D. et al. Shale Brittleness Index Based on the Energy Evolution Theory and Evaluation with Logging Data: A Case Study of the Guandong Block. ACS Omega 2020, 5, 22, 13164-13175.

[Comment 3]: In Figure 4, the division of the curve into three parts in the Full stress-strain curve in three modulus form appears to be hasty. It is recommended that the authors provide additional explanations and evidence regarding the feasibility of this theoretical approach.

[Reply 3]: Thank you for your kind suggestions. Figure 4 is the theoretical representation of Figure 1, as described in Question 2, which is divided into energy accumulation stage, energy dissipation stage, and energy release stage. Many scholars have adopted simplified stress-strain curve models for verification. This is a theoretical model for our research, in which three sections E, D, and M lay the foundation for deriving a mathematical model for calculating the brittleness index. We supplement and improve Figure 4 to better demonstrate the energy evolution of rocks and the stress-strain relationship. At the same time, we also added relevant explanations about Figure 4.

[Changes]: Please see Line 226-241 on Page 7 in the "Tracked changes file".

[Comment 4]: The lines in Figure 5 are overlapping and appear to be visually confusing, which makes the logic unclear. I suggest redrawing the figure to improve its clarity.

[Reply 4]: Thank you for your comments. Figure 5 has been redrawn and displayed in the revised manuscript. Related description has been added.

[Changes]: Please see Line 374-384 on Page 12 in the "Tracked changes file".

[Comment 5]: The relationship between the content presented in Figure 7 and the accompanying text stating ‘Figure 7 shows the relationship between elastic/yield/post-peak modulus and well 349 depth...’ (lines 347-358) is not clearly evident. It is recommended that the authors revise and refine this section to ensure better alignment between the visual content and the accompanying text.

 [Reply 5]: Thank you for your comments. Due to negligence in mixing the ‘weakening module’ and ‘post peak module’, we have revised the relevant description of Figure 7 to ensure better match between the article and the images. Meanwhile, we also adjusted the positions of Figures (b) and (c). Please refer to the revised manuscript for revisions.

[Changes]: Please see Line 393-406 on Page 13-14 in the "Tracked changes file".

[Comment 6]: Some related papers (e.g., Journal of Cleaner Production 354 (2022) 131724) are suggested to cited.

 [Reply 6]: Thank you for your comments. The relevant literatures have been cited in the appropriate position of the article, as shown in the revised manuscript.

Reference:

[1] Zhang., Y.; Li., C.; Jiang., Y.; Sun., L.; Zhao., R.; Yan., K.; Wang., W. Accurate prediction of water quality in urban drainage network with integrated EMD-LSTM model. Journal of Cleaner Production, 2022, 354: 131724.

[2] Fu, P.; Settgast, R. R.; Hao, Y.; Morris, J. P.; Ryerson, F. J. The influence of hydraulic fracturing on carbon storage performance. Journal of Geophysical Research: Solid Earth, 2017, 122: 9931–9949.

[2] Wang., M.; Li., L.; Peng., X.; Hu., Y.; Wang., X.; Luo, Y.; Yu., Peng. Influence of stress redistribution and fracture orientation on fracture permeability under consideration of surrounding rock in underground gas storage. Energy Reports. 2022, 8: 6563-6575.

[3] Xue., W.; Wang., Yi.; Chen., Zhe.; Liu., H. An integrated model with stable numerical methods for fractured underground gas storage. Journal of Cleaner Production. 2023, 393: 136268.

[4] Zhang., Y.; Zhang., L.; He., J.; Zhang., H.; Zhang., X.; Liu., X. Fracability Evaluation Method of a Fractured-Vuggy Carbonate Reservoir in the Shunbei Block. ACS Omega, 2023, 8(17): 15810-15818.

[5] Zeng., F.; Gong., G.; Zhang., Y.; Guo., J.; Jiang., J.; Hu., D.; Chen., Z. Fracability evaluation of shale reservoirs considering rock brittleness, fracture toughness, and hydraulic fracturing-induced effects. Geoenergy Science and Engineering, 2023, 229 : 212069.

[6] Dou, L.; Zuo, X.; Qu, L.; Xiao, Y.; Bi, G.; Wang, R.; Zhang, M. A New Method of Quantitatively Evaluating Fracability of Tight Sandstone Reservoirs Using Geomechanics Characteristics and In Situ Stress Field. Processes, 2022, 10: 1040.

 [Changes]: Please see Line 42-59 on Page 2, Line 499-512 on Page 20 in the "Tracked changes file".

[Comment 7]: Many grammar issues in the text need to be corrected. For example, in the fourth part, ‘The key energy’ (lines 433) should be corrected. The sentence ‘which is more unfavorable for reservoir fracturing’ (lines 442) is grammatically correct and there are no language errors. However, it may be unclear what "which" is referring to without the context of the previous sentence.

[Reply 7]: Thank you for your comments. We have corrected the related grammar error, and “which” refers to the content “As the depth of the reservoir increases, both the elastic modulus, yield modulus and post-peak modulus decrease, resulting in a decrease in reservoir brittleness and fracability”.

[Changes]: Please see Line 476-497 on Page 20 in the "Tracked changes file".

******************************************************************************************************************************************

Close

Reviewer 2 Report

Comments and Suggestions for Authors

The development of effective technologies for gas storage is very important for the exploitation of resources. The authors of the paper “A Logging Data Based Method for Evaluating the Fracability of a Gas Storage in Eastern China” have developed a mathematical model for the prediction of stress distribution and possible fracture in underground storage of natural gas. The authors have presented a significant number of calculated results. However, some points of the paper are needed to be explained in more detail accordingly following comments:

1.                  The most of the analyzed references in the Introduction part is too old. The authors should analyzed last papers about fracture mechanics application for underground storage of natural gas.

2.                  The authors use for analysis stress-strain curve obtained for uniaxial compression. However, in real construction another stress conditions may be realized. What is the stress state in the real construction of the underground storage of natural gas? Most probable situation that stress condition is all-round (three axial) compression. How the constructed model may be apply in this case?

3.                  The main problem of the paper is absence of the experimental approvement of the developed index. The authors should provide information how their findings may be approved experimentally. It is recommended also to check constructed models using finite element simulation of the real construction.

4.        The presented stress-strain scheme is too simplified. The crack propagation after the σB proceeds spontaneously without application any additional force due to brittle fracture. In my opinion, it is not correct to include this range to britlleness index.

5.        Accordingly Figure 1 elastic modulus has a value about 20 GPa. However, in Line 351 the value is  mainly distributed between 10MPa and 40MPa. Please, check the correctness of the units.

6.        Figure 7: units of the elastic modulus are GPa not MPa. Check units for other stresses.(c) and (b) figures have incorrect captions.

7.        It is recommended to decrease the number of digits in the properties values in Conclusion part and to add the units to elastic modulus.

Author Response

Responses to Technical Editor 2: 

The development of effective technologies for gas storage is very important for the exploitation of resources. The authors of the paper “A Logging Data Based Method for Evaluating the Fracability of a Gas Storage in Eastern China” have developed a mathematical model for the prediction of stress distribution and possible fracture in underground storage of natural gas. The authors have presented a significant number of calculated results. However, some points of the paper are needed to be explained in more detail accordingly following comments:

[Comment 1]: The most of the analyzed references in the Introduction part is too old. The authors should analyze last papers about fracture mechanics application for underground storage of natural gas.

 [Reply 1]: Thank you for your careful review. The recent literature has been added in the introduction. Relevant literature on the impact of fracturing on underground natural gas storage has also been cited and explained.

References:

[1] Fu, P.; Settgast, R. R.; Hao, Y.; Morris, J. P.; Ryerson, F. J. The influence of hydraulic fracturing on carbon storage performance. Journal of Geophysical Research: Solid Earth, 2017, 122: 9931–9949.

[2] Wang., M.; Li., L.; Peng., X.; Hu., Y.; Wang., X.; Luo, Y.; Yu., Peng. Influence of stress redistribution and fracture orientation on fracture permeability under consideration of surrounding rock in underground gas storage. Energy Reports 8 (2022) 6563–6575.

[3] Xue., W.; Wang., Yi.; Zhe., Chen.; Hang., Liu. An integrated model with stable numerical methods for fractured underground gas storage. Journal of Cleaner Production 393 (2023) 136268.

[4] Zhang., Y.; Zhang., L.; He., J.; Zhang., H.; Zhang., X.; Liu., X. Fracability Evaluation Method of a Fractured-Vuggy Carbonate Reservoir in the Shunbei Block. ACS Omega, 2023, 8(17): 15810–15818.

[5] Zeng., F.; Gong., G.; Zhang., Y.; Guo., J.; Jiang., J.; Hu., D.; Chen., Z. Fracability evaluation of shale reservoirs considering rock brittleness, fracture toughness, and hydraulic fracturing-induced effects. Geoenergy Science and Engineering, 2023, 229 : 212069.

[6] Dou, L.; Zuo, X.; Qu, L.; Xiao, Y.; Bi, G.; Wang, R.; Zhang, M. A New Method of Quantitatively Evaluating Fracability of Tight Sandstone Reservoirs Using Geomechanics Characteristics and In Situ Stress Field. Processes, 2022, 10: 1040.

 [Changes]: Please see Line 43-59 on Page 2; Line 476-497 on Page 20 in the "Tracked changes file".

[Comment 2]: The authors use for analysis stress-strain curve obtained for uniaxial compression. However, in real construction another stress conditions may be realized. What is the stress state in the real construction of the underground storage of natural gas? Most probable situation that stress condition is all-round (three axial) compression. How the constructed model may be applied in this case?

 [Reply 2]: The stress experienced by underground natural gas storage is triaxial, and under most real conditions, the stress conditions are also triaxial. Triaxial compression changes the strength of rocks with changes in confining pressure, and the same rock will have different brittleness indices due to different confining pressures. The brittleness index in this article simply reflects the properties of a certain rock and does not consider the influence of external factors such as confining pressure. So we usually use uniaxial compression experiments, but if the depth of the formation is specified, triaxial compression experiments can also be used.

In order to compensate for the impact of geological environment, we have also taken into account the ground stress environment in the fracability index, and integrated the brittleness index and ground stress difference coefficient, which makes the consideration more comprehensive.

[Comment 3]: The main problem of the paper is absence of the experimental approvement of the developed index. The authors should provide information how their findings may be approved experimentally. It is recommended also to check constructed models using finite element simulation of the real construction.

[Reply 3]: Thank you for your useful comments. We can obtain the complete stress-strain curve of rocks through uniaxial compression tests, and then calculate the elastic modulus, yield modulus, and post peak modulus from the stress-strain curve. Li et al. (2019) have conducted related experiment based on energy method to calculate the brittleness on coal. In future work, we can also conduct relevant experimental research based on gas storage facilities.

The fracability evaluation model in this paper is an analytical model before fracturing, and the cracks generated by finite element simulation are after fracturing, so finite element simulation of this model is not applicable.

[Changes]: Please see Line 226-228 on Page 7 in the "Tracked changes file".

Reference:

[1] Li., Y.; Long., M.; Zuo., L.; Li., W.; Zhao., W. Brittleness evaluation of coal based on statistical damage and energy evolution theory, Journal of Petroleum Science and Engineering, 2019, 172: 753-763.

[Comment 4]: The presented stress-strain scheme is too simplified. The crack propagation after the σB proceeds spontaneously without application any additional force due to brittle fracture. In my opinion, it is not correct to include this range to brittleness index.

 [Reply 4]: Thank you for your helpful suggestions. When the rock material reaches its peak strength σB, the elastic energy stored inside the rock sample is not sufficient to maintain complete failure of the sample, so additional energy needs to be provided externally to support this process. Under mechanical experimental conditions, this portion of energy is provided by the continued loading of the testing machine. When the stress drops to the residual strength σC, the elastic energy and additional energy are converted into the fracture energy required to maintain the failure of the specimen, while there is still some residual elastic performance inside the specimen. This article uses the energy method to calculate the brittleness index of rocks, focusing on the brittleness caused by rock structure and composition. Therefore, the peak strength σB has an impact on the rock structure, which in turn affects the brittleness index.

[Comment 5]: Accordingly Figure 1 elastic modulus has a value about 20 GPa. However, in Line 351 the value is mainly distributed between 10MPa and 40MPa. Please, check the correctness of the units.

[Reply 5]: Thank you for your useful comments. We have checked the unit of elastic modulus in the article and changed MPa to GPa.

[Changes]: Please see Line 468-471 on Page 19 in the "Tracked changes file".

[Comment 6]: Figure 7: units of the elastic modulus are GPa not MPa. Check units for other stresses. (c) and (b) figures have incorrect captions.

 [Reply 6]: Thank you for your useful comments. We have made relevant modifications to the modulus unit, and the positions of figures (b) and (c) have also been swapped.

[Changes]: Please see Line 393-406 on Page 13-14; Line 468-471 on Page 19 in the "Tracked changes file".

[Comment 7]:  It is recommended to decrease the number of digits in the property values in Conclusion part and to add the units to elastic modulus.

[Reply 7]: Thank you for your useful comments. We have reduced the number of attribute values in the conclusion section to 3 decimal places and added units, GPa, to the elastic modulus.

[Changes]: Please see Line 493-497 on Page 20 in the "Tracked changes file".

******************************************************************************************************************************************

Close

Round 2

Reviewer 2 Report

Comments and Suggestions for Authors

The authors have answered previous comments and improved the manuscript. The paper may be accepted for publication.